# Planar Waveguide-Based Fiber Spectrum Analyzer Mountable to Commercial Camera

Xinhong Jiang 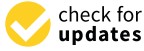 and Ziyang Zhang *

Laboratory of Photonic Integration, School of Engineering, Westlake University, Hangzhou 310024, China; jiangxinhong@westlake.edu.cn
* Correspondence: zhangziyang@westlake.edu.cn

**Abstract:** We present the design of a planar spectrometer that separates the wavelength channels from an input fiber and focuses the spectral lines onto a camera without any free-space optical elements. The geometric arrangements of the waveguides to achieve different spectroscopic parameters are explained in detail, allowing adjustable focal lengths, high spectral resolution, and broad free spectral range. The optical chip is fabricated on a low-cost polymer platform as proof of concept. The optical spectrum of a multiwavelength laser is measured by the proposed device, and the result is in good agreement with a commercial optical spectrum analyzer. The large focal depth of the chip allows an optical assembly of much relaxed alignment accuracy. We demonstrate a tube design to encapsulate the chip fixed with the input fiber. The assembly is then mounted to a commercial camera with standard C-mount threading as a convenient fiber spectrum analyzer without customized detectors and circuits. Our design may provide a low-cost and versatile solution for the development of compact spectroscopic equipment.

**Keywords:** spectroscopy; integrated optics; planar lens



## 1. Introduction

Spectroscopy is a fundamental tool for the investigation of material properties through the interaction with electromagnetic radiation. It can be applied in a wide range of research areas, from structures of atoms and molecules to space exploration [1–4]. Compact, lightweight, and integrated spectrometers are desired for many applications as they can be fabricated at a relatively low cost, and the technology for system assembly can be readily transferred from the development of modules in optical communication. Arrayed waveguide gratings (AWGs) have been widely used in the field of spectrum analysis, thanks to its compact footprint and high resolution [5–8]. To detect the output lights of an AWG, photodiodes (PDs) can be integrated by means of hybrid [9,10] or monolithic integration [11–13]. In hybrid integration, a PD array can be placed on top of the chip via an integrated 45-degree mirror, which requires an alignment precision on the micrometer scale. In monolithic integration, Ge or InGaAs PDs can be integrated on silicon or InP platforms, respectively. However, in both cases, customized electronic circuits should be made to relay and process the photocurrent, which may increase the system cost and raise challenge in reliability.

Alternatively, the outputs of an AWG can be imaged onto line array detectors or cameras [7,14]. The output waveguides of the AWGs must then be cut away along the output facet of the second slab. Since light is already focused on this facet, a free-space lens is needed to refocus the beam to the detectors. In addition, the curvature of the output facet causes defocus aberration for off-axis beams, which decreases the resolution.

The adoption of the free-space elements inevitably enlarges and complicates the spectrometer system. To solve this problem, the beam focusing function should be preferably integrated in the optical chip itself. This can be achieved by adding additional length

difference to the waveguide array, according to the phase modulation of a convex lens. In our previous work, we demonstrated the possibility of this technology and presented the concept of a new device called waveguide spectral lens (WSL). Two devices have been demonstrated with a full width at half maximum (FWHM) of 0.63 and 0.42 nm and a free spectral range (FSR) of ~62 and ~59 nm, respectively [15].

Spectrometers with high resolutions and broad FSRs are required for different applications. For surveys in astronomy and Raman analysis of materials, the spectrometer should have a broad working range, and the FSR should be sufficiently wide to limit the need of using a cross disperser. On the other hand, high-resolution spectrometers are desired for applications such as wavelength characterization of lasers and precise atomic emission lines. Furthermore, to allow flexibility in building the spectrometer system, the location of the camera should be favorably adjustable. Therefore, the focal length of the WSL should also be adaptable in the design.

In this work, we perform a comprehensive study of the design, fabrication, and testing of WSLs and develop the technology to mount a WSL to a commercial camera conveniently with minimal cost. By designing the length difference of the waveguide array, WSLs with different focal lengths, FWHMs, and FSRs can be realized. As the one-arc structure used in [15] is not suitable for WSLs with small length difference (i.e., large FSR), a three-arc structure is proposed to solve this problem. Detailed geometrical arrangements with equations and diagrams are revealed. In the experiment, the beam focusing function of a planar waveguide lens without dispersion function is demonstrated first, followed by the demonstration of WSLs with different focal lengths, FWHMs, and FSRs. The spectrum of a multiwavelength laser (MWL) is measured by a chosen WSL and compared with a standard commercial optical spectral analyzer (OSA). Finally, an adjustable tube is designed for mounting the chip to a commercial camera as a low-cost, efficient, and convenient fiber spectrum analyzer.

The paper is organized as follows: Section 2 describes the device structure and operation principle. In Section 3, the structural parameters of a waveguide lens and WSLs with different focal lengths, FWHMs, and FSRs are calculated. Section 4 presents the device fabrication and testing results. Section 5 reveals the integration technology and demonstrates the mounting of a tube device to a commercial camera. Section 6 makes the conclusion and lays out plans for further development.

## 2. Device Structure and Operation Principle
### 2.1. Device Structure and Waveguide Length Difference

Figure 1 shows the schematics of WSLs with the waveguide array adopting the one-arc and three-arc structures to allow flexibility in setting length differences in the waveguide array. In general, a WSL chip consists of an input waveguide, a beam broadening area (BBA), and a waveguide array. The input waveguide is placed at the center of the BBA. The starting points of the waveguides in the array are placed along the perimeter of the BBA in order to tap out the light wavefront with equal phase. The lengths of the waveguides are designed to integrate both wavelength separation and beam focusing functions. The output waveguides are cut open along a line perpendicular to the equally spaced parallel waveguides. The facets form an array of emitters, and the joint far field/diffraction pattern is captured by a camera placed at a given distance much larger than the wavelength $\lambda$.

The length of the $i$-th waveguide is defined as $l_i$. The length difference between the $i$-th waveguide and the first waveguide is designed as:

$$l_i - l_1 = P_i - P_1 + (i - 1)\Delta L, \ (i = 1, \dots, N),$$ (1)

$$P_i = \left( f - \sqrt{f^2 + D_i{}^2} \right) / n_{\text{eff}},$$ (2)

$$\Delta L = 1 / \left( n_{\text{eff}} / \lambda_{\text{start}} - n_{\text{eff}} / \lambda_{\text{stop}} \right),$$ (3)

$$\text{FSR} = \lambda_{\text{stop}} - \lambda_{\text{start}} \approx \lambda^2 / (n_{\text{eff}} \Delta L),$$ (4)

where $P_i$ is the required relative length in the *i*-th waveguide for the beam focusing function [16]. $\Delta L$ is the uniform length difference of adjacent waveguides for wavelength separation. $N$ is the number of waveguides. $f$ is the focal length. $D_i$ is the distance between the *i*-th waveguide output facet and the center of the whole output facet. $n_{\mathrm{eff}}$ is the effective index of the waveguides. $\lambda_{\mathrm{start}}$ and $\lambda_{\mathrm{stop}}$ are the start and stop wavelengths within one FSR.

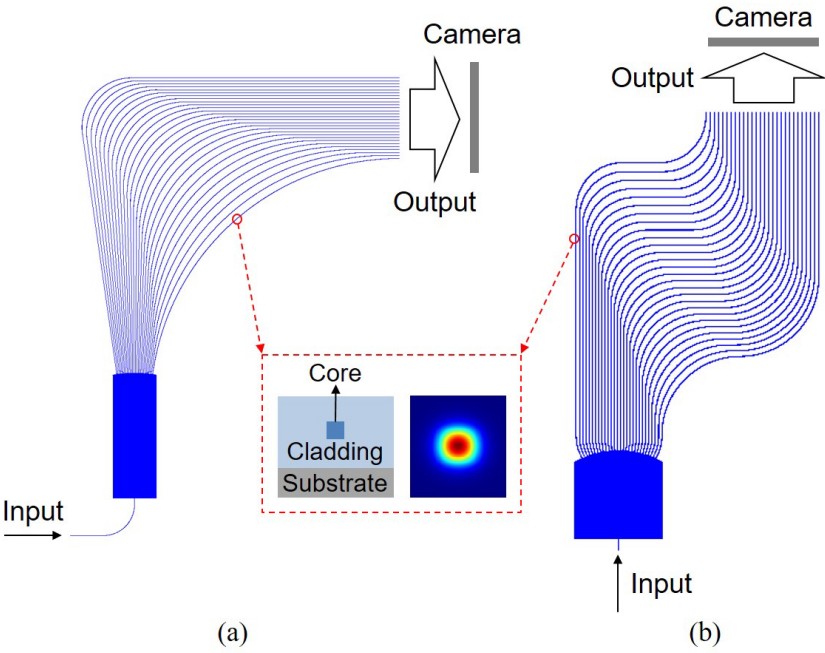

**Figure 1.** Schematics of WSLs with the waveguide array adopting the (**a**) one-arc and (**b**) three-arc structures to allow flexibility in setting length differences in the waveguide array.

Figure 2 illustrates the wavelength separation and beam focusing functions of a WSL. The deflection angle is determined by the uniform phase difference between adjacent waveguides. An additional phase difference is designed according to the phase modulation of a convex lens and added to the deflected equiphase plane. Lights from the center of the output facet (point O) and the output facet of the *i*-th waveguide (point I) arrive at points O′ and I′ in the equiphase plane, respectively, and intersect with the same phase at point F. The line O′F is perpendicular to the equiphase plane. The optical path length difference of O′F and I′F should be equal to the optical path difference $n_{\mathrm{eff}}P_i$ generated by the waveguide array for the focusing effect. According to Equation (2), the length difference $\Delta P$ of O′F and I′F can be written as

$$\Delta P = f_i - \sqrt{f_i{}^2 + (D_i\cos(\theta))^2} = f - \sqrt{f^2 + D_i{}^2}, \tag{5}$$

where $f_i$ is the length of O′F. $\theta$ is the deflection angle of the far field, which satisfies the following equation [17]:

$$\theta = \arcsin(\Delta\varphi/(kd)), \tag{6}$$

where $\Delta\varphi = n_{\mathrm{eff}}k\Delta L$ is the uniform phase difference for the grating effect, with $k$ denoting the wavenumber. $d$ is the waveguide spacing at the output facet of the waveguide array. In the calculation of the deflection angle, $\Delta\varphi$ should be normalized in the range of $-\pi$ to $\pi$.

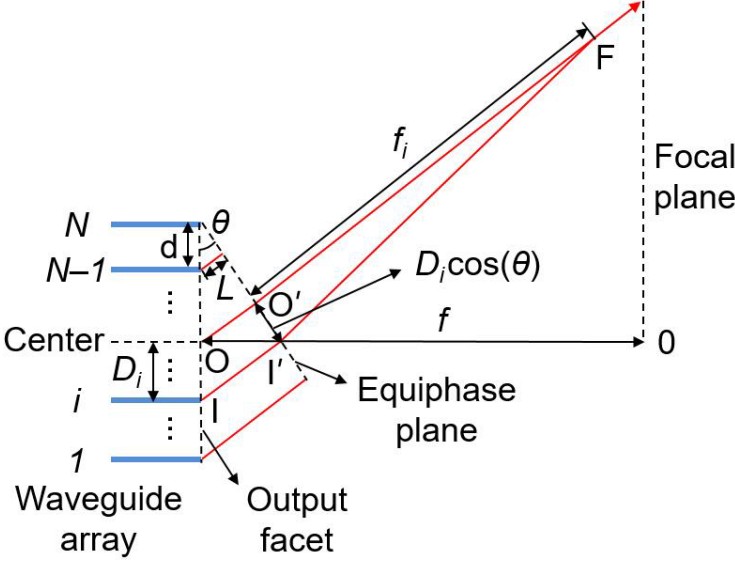

**Figure 2.** Illustration of the wavelength separation and beam focusing functions of a waveguide array.

According to Equation (5), $f_i$ can be calculated as

$$f_i = \frac{\left(f - \sqrt{f^2 + D_i^2}\right)^2 - (D_i \cos(\theta))^2}{2(f - \sqrt{f^2 + D_i^2})}. \tag{7}$$

It can be seen that $f_i$ changes with $\theta$ and $D_i$. Under paraxial approximation, when the deflection angle $\theta$ is small, $f_i$ is close to $f$, and light from different waveguide facets can be focused.

The field of view (FOV) of the far field of the waveguide array is given by [18]:

$$\text{FOV} = 2\arcsin(\lambda/(2d)). \tag{8}$$

The beam width $\Delta\theta_{\text{FWHM}}$ of the far field can be written as [17]

$$\Delta\theta_{\text{FWHM}} \approx \frac{0.886\lambda}{Nd\cos(\theta)}. \tag{9}$$

According to Equations (8) and (9), the ratio of FOV and $\Delta\theta_{\text{FWHM}}$ can be obtained, which is equal to the ratio of FSR and FWHM. When $\theta$ is close to zero, the ratio can be written as

$$\text{FSR/FWHM} \approx 1.13N. \tag{10}$$

### 2.2. Calculation of Structure Parameters

2.2.1. Detailed Geometric Structures of WSLs

To illustrate the structural parameters of WSLs, two geometric structures of WSLs are plotted in Figure 3. The geometric structure with one arc in each waveguide in Figure 3a is suitable for designing WSLs with large length differences, corresponding to small FSRs. However, if the one-arc structure is used for a WSL with a small length difference, the waveguide spacing of the waveguide array will be reduced, which increases the crosstalk between waveguides. To solve this problem, the geometric structure with three arcs in each waveguide is proposed for designing WSLs with short length differences, corresponding to large FSRs, as sketched in Figure 3b.

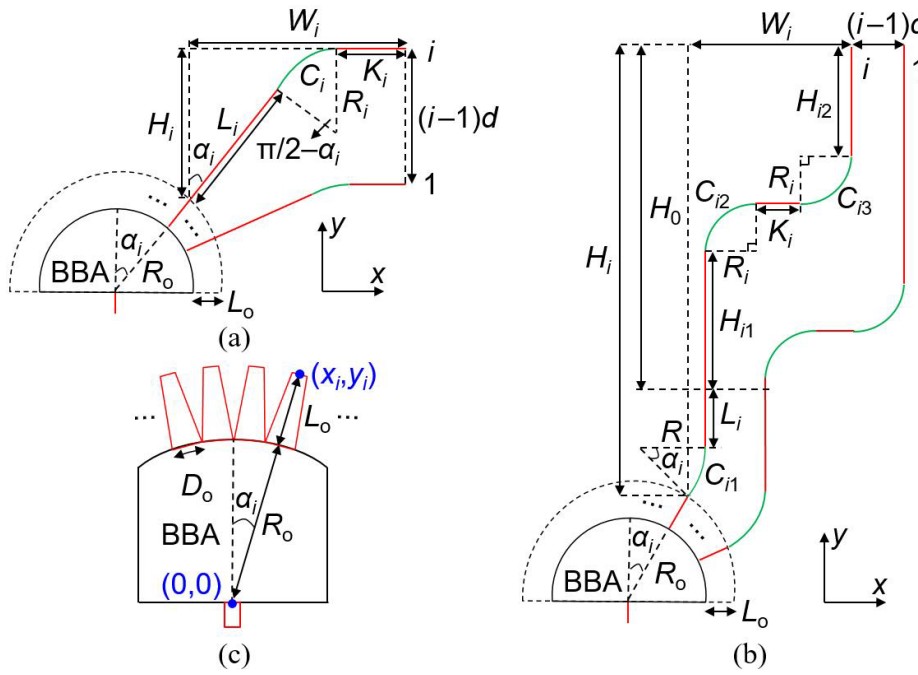

**Figure 3.** (**a**,**b**) Geometric structures with (**a**) one arc and (**b**) three arcs in each waveguide for designing WSLs with large and small length differences, respectively. (**c**) A BBA with one input waveguide and multiple output tapers.

As shown in Figure 3c, to tap out the wavefront of the broadened incoming light more efficiently, the first segment of each waveguide in the waveguide array is arranged in a taper form. $x_i$ and $y_i$ are defined as the $x$ and $y$ coordinates of the ending point of the taper in the $i$-th waveguide, which can be calculated as

$$x_i = (R_o + L_o) \sin(\alpha_i), \tag{11}$$

$$y_i = (R_o + L_o) \cos(\alpha_i), \tag{12}$$

$$\alpha_i = ((N+1)/2 - i) D_o / R_o, \tag{13}$$

where $L_o$ is the taper length of each waveguide. $R_o$ is the radius of the BBA. $D_o$ is the length of the arc connecting adjacent waveguides at the output facet of the BBA. The front facet of the taper is the chord of an arc with a length of $D_o$. $\alpha_i$ is the angle between the central line of the taper and the $y$ axis, which is defined as negative or positive values for waveguides connecting with the left or right parts of the BBA, respectively.

In Figure 3a, the $i$-th waveguide in the array consists of a taper segment $L_o$, a straight segment $L_i$, an arc segment $C_i$, and the output straight segment $K_i$. The length of the $i$-th waveguide is $l_i = L_o + L_i + C_i + K_i$. These lengths are designed to satisfy the length difference requirement in Equation (1). $R_i$ and $\pi/2$—$\alpha_i$ are the radius and angle of the arc segment $C_i$ in the $i$-th waveguide, respectively.

In Figure 3b, the $i$-th waveguide in the array consists of a taper segment $L_o$, an arc segment $C_{i1}$, a straight segment $L_i$, a straight segment $H_{i1}$, an arc segment $C_{i2}$, a straight segment $K_i$, an arc segment $C_{i3}$, and the output straight segment $H_{i2}$. The length of the $i$-th waveguide is $l_i = L_o + C_{i1} + L_i + C_{i2} + K_i + C_{i3} + H_0 - 2R_i$, where $H_0 = H_{i1} + 2R_i + H_{i2}$. $R$ and $|\alpha_i|$ are the radius and angle of the first arc. The radius and angle of both the second and third arcs are $R_i$ and $\pi/2$, respectively.

### 2.2.2. Derivation for the Formulas of Structural Parameters

Three structural parameters, $L_i$, $K_i$, and $R_i$, should be calculated according to the three constraint conditions in Figure 3a,b.

The structural parameters of the $i$-th waveguide in Figure 3a are subject to three constraint conditions: (1) the length difference $l_i - l_1$ should satisfy Equation (1), (2) the $x$ values of the ending points of all waveguides are the same, and (3) the $y$ value of the ending point of the $i$-th waveguide is $(i - 1)d$ larger than that of the first waveguide. They can be written as:

$$L_i + (\pi/2 - \alpha_i)R_i + K_i - (L_1 + (\pi/2 - \alpha_1)R_1 + K_1) = l_i - l_1, \tag{14}$$

$$L_i \sin(\alpha_i) + R_i \cos(\alpha_i) + K_i - (L_1 \sin(\alpha_1) + R_1 \cos(\alpha_1) + K_1) = x_1 - x_i, \tag{15}$$

$$L_i \cos(\alpha_i) + R_i(1 - \sin(\alpha_i)) - (L_1 \cos(\alpha_1) + R_1(1 - \sin(\alpha_1))) = y_1 - y_i + (i - 1)d. \tag{16}$$

By solving Equations (14)–(16), the structural parameters can be calculated as:

$$R_i = b/c, \tag{17}$$

$$L_i = (a - R_i(\cos(\alpha_i) - (\pi/2 - \alpha_i)))/(\sin(\alpha_i) - 1), \tag{18}$$

$$K_i = l_i - l_1 + (L_1 + (\pi/2 - \alpha_1)R_1 + K_1) - (L_i + (\pi/2 - \alpha_i)R_i), \tag{19}$$

where

$$a = x_1 - x_i + (L_1 \sin(\alpha_1) + R_1 \cos(\alpha_1) + K_1) - (l_i - l_1) - (L_1 + (\pi/2 - \alpha_1)R_1 + K_1), \tag{20}$$

$$b = (i - 1)d + y_1 - y_i - a/(\sin(\alpha_i) - 1)\cos(\alpha_i) + (L_1 \cos(\alpha_1) + R_1(1 - \sin(\alpha_1))), \tag{21}$$

$$c = -(\cos(\alpha_i) - (\pi/2 - \alpha_i))/(\sin(\alpha_i) - 1)\cos(\alpha_i) + (1 - \sin(\alpha_i)). \tag{22}$$

Similarly, the structural parameters of the $i$-th waveguide in Figure 3b are subject to three constraint conditions: (1) the length difference $l_i - l_1$ should satisfy Equation (1), (2) the $x$ value of the ending point of the $i$-th waveguide is $(i - 1)d$ smaller than that of the first waveguide, and (3) the $y$ values of the ending points of all waveguides are the same. They can be written as:

$$R|\alpha_i| + L_i + \pi R_i + K_i - 2R_i - (R|\alpha_1| + L_1 + \pi R_1 + K_1 - 2R_1) = l_i - l_1, \tag{23}$$

$$\begin{aligned} R(1 - \cos(\alpha_i)) + 2R_i + K_i - (R(1 - \cos(\alpha_1)) + 2R_1 + K_1) = x_1 - x_i - (i - 1)d, \ (\alpha_i \geq 0), \\ -R(1 - \cos(\alpha_i)) + 2R_i + K_i - (R(1 - \cos(\alpha_1)) + 2R_1 + K_1) = x_1 - x_i - (i - 1)d, \ (\alpha_i < 0), \end{aligned} \tag{24}$$

$$R \sin(|\alpha_i|) + L_i - (R \sin(\alpha_1) + L_1) = y_1 - y_i. \tag{25}$$

By solving Equations (23)–(25), the structural parameters can be calculated as:

$$L_i = y_1 - y_i + (R \sin(\alpha_1) + L_1) - R \sin(|\alpha_i|), \tag{26}$$

$$\begin{aligned} R_i = (x_1 - x_i - (i - 1)d + (R(1 - \cos(\alpha_1)) + 2R_1 + K_1) - (R(1 - \cos(\alpha_i)) \\ + l_i - l_1 + (L_1 + R\alpha_1 + (\pi - 2)R_1 + K_1) - (L_i + R|\alpha_i|)))/(4 - \pi), \ (\alpha_i \geq 0), \\ R_i = (x_1 - x_i - (i - 1)d + (R(1 - \cos(\alpha_1)) + 2R_1 + K_1) - (-R(1 - \cos(\alpha_i)) \\ + l_i - l_1 + (L_1 + R\alpha_1 + (\pi - 2)R_1 + K_1) - (L_i + R|\alpha_i|)))/(4 - \pi), \ (\alpha_i < 0), \end{aligned} \tag{27}$$

$$K_i = l_i - l_1 + (L_1 + R\alpha_1 + (\pi - 2)R_1 + K_1) - (L_i + R|\alpha_i| + (\pi - 2)R_i). \tag{28}$$

### 2.2.3. Calculation of Structural Parameters Based on the Formulas

To obtain the structural parameters, the required FSR is used to calculate the length difference $\Delta L$ according to Equations (3) and (4). Then the sweep ranges of the parameters $R_o$, $D_o$, $L_1$, $K_1$, and $R_1$ are set according to the selected material platform. For each group of $R_o$, $D_o$, $L_1$, $K_1$, and $R_1$, the variables $L_i$, $K_i$, and $R_i$ can be calculated according to Equations (17)–(19) or Equations (26)–(28). The amplitude and phase of each waveguide at the output facet of the BBA are calculated by a Rayleigh–Sommerfeld diffraction integral [19] from the near-field profile of the input waveguide. Similarly, the far fields at a chosen focal length $f$ are calculated again by Rayleigh–Sommerfeld diffraction integral collectively from the near fields at the output facets, for each group of structural parameters.

The structural parameters are selected according to the device footprint, the distance between adjacent waveguides, and the required FWHM. The waveguide spacing $d$ at the output facet is designed according to the width of the far field $\Delta D$ of an FSR on the focal plane, which should be smaller than the width of the camera sensor; thus, the far field in an entire FSR can be captured by the camera. $D$ is the position in the dispersion direction on the focal plane. $\Delta D$ can be approximately calculated as

$$\Delta D = 2f \tan(\text{FOV}/2). \tag{29}$$

The camera used in the experiment has $640 \times 512$ pixels with a pitch of 20 μm. For the polymer waveguide platform, the core and cladding material indexes at 1550 nm are ~1.48 and ~1.45, respectively. The cross section of the waveguide is 3 μm $\times$ 3 μm. The effective index of the waveguide at 1550 nm is 1.4625, and the mode field diameter is ~4 μm. In the calculation of the structure parameters, the distances between adjacent waveguides are designed to be larger than 12 μm to suppress the crosstalk between waveguides for the chosen waveguide platform. The minimum length of the output straight segments is set to 20 μm to facilitate chip dicing along the output facet. The taper lengths $L_o$ are selected according to the width variations of the taper. The minimum gap between the tapers is increased to larger than 1 μm by extending the BBA along its central line, in consideration of the fabrication limitation of a standard contact photolithography.

### 2.3. Calculation of Focal Depth

The focal depth describes the ability of the WSL chip to render a sharp spectral line even when the camera location deviates from the designed focal plane. We set the threshold as 10% widening of the minimal FWHM of the spectral line and define the focal depth as the allowed distance between the two threshold planes in front of and after the focal plane.

The FWHM of the far field is proportional to $|\Delta z|(N{-}1)d/f$, where $|\Delta z|$ is the distance between the sensor plane and the focal plane. $(N{-}1)d$ is the output facet width of the waveguide array [15]. The WSL chip can be designed to have a focal length on the centimeter scale and an output facet width on the millimeter scale. Therefore, the focal depth can reach millimeter scale, as discussed in our previous work [15]. A large focal depth relaxes the alignment accuracy between the WSL chip and the camera sensor. The WSL chip can then be mounted to a camera without requiring high alignment accuracy. A tube is designed to encapsulate the WSL chip and mounted to a commercial camera, as demonstrated in Section 5.

### 3. Device Function Definition

Based on the calculation method in Section 2, a waveguide lens without grating function and WSLs with different parameters can be designed. The waveguide lens is used to demonstrate the beam focusing function. The WSLs have different focal lengths, FWHMs, and FSRs. The three-arc structure in Figure 3b is used to design the waveguide lens (WG lens) and the WSL with a small length difference (WSL4). The one-arc structure is used to design WSLs with large length differences (WSLs 1–3). The designed parameters (i.e., number of waveguides ($N$), free spectral range (FSR), uniform length difference between adjacent waveguides ($\Delta L$), focal length ($f$), waveguide spacing at the output facet of the waveguide array ($d$), radius of the BBA ($R_o$), length of the arc connecting two adjacent waveguides at the output facet of the BBA ($D_o$), length of the taper ($L_o$), and parameters $R$, $L_1$, $K_1$, $R_1$ defined in Figure 3) of the devices are given in Table 1. The calculated $\Delta D$, the FWHM, and the focal depth are also listed in Table 1. The calculated focal depths are on the millimeter scale, which means that the camera can be aligned with the WSL chip without high alignment accuracy.

The simulated FWHMs of WSL1–3 are close to the values calculated from Equation (10), but the deviation is relatively large for WSL4, which can be attributed to the small amplitudes in the waveguides at the two sides of the waveguide array. Figure 4 shows the normalized amplitudes at the output facets of the BBAs in the WSLs. For WSL4, the

normalized amplitudes in the waveguides at the two sides are small (~0.02) and have little contribution to the far field, similar to the effect of decreasing $N$. However, the truncation losses at the output facet of the BBAs of WSL1–3 are relatively large since less output power of the BBA is received by the waveguide array.

**Table 1.** Designed and measured parameters of waveguide lens and WSLs.

| Parameter | WG Lens | WSL1 | WSL2 | WSL3 | WSL4 |
|---|---|---|---|---|---|
| $N$ | 220 | 120 | 220 | 60 | 120 |
| FSR (nm) | ~ [1] | 64 | 31 | 4.5 | 112 |
| $\Delta L$ (µm) | 0 | 25.7 | 53.0 | 365.1 | 14.6 |
| $f$ (cm) | 10 | 5 | 10 | 10 | 10 |
| $d$ (µm) | 18 | 16.5 | 13.5 | 66 | 33 |
| $R_o$ (µm) | 3000 | 6000 | 6000 | 6000 | 6000 |
| $D_o$ (µm) | 12 | 8 | 8 | 48 | 36 |
| $L_o$ (µm) | 200 | 200 | 200 | 600 | 500 |
| $R$ (µm) | 1500 | ~ | ~ | ~ | 1500 |
| $L_1$ (µm) | 0 | 0 | 0 | 4000 | 500 |
| $K_1$ (µm) | 0 | 20 | 20 | 20 | 0 |
| $R_1$ (µm) | 2700 | 3000 | 13,500 | 21,000 | 2800 |
| Calculated $\Delta D$ (µm) | ~ | 4702.2 | 11,500.4 | 2348.6 | 4698.3 |
| Simulated FWHM (nm) | ~ | 0.50 | 0.14 | 0.09 | 1.47 |
| Focal depth (mm) | 4.30 | 4.51 | 7.48 | 4.17 | 4.9 |
| Measured FWHM (nm) | ~ | 0.74 | 0.16 | 0.15 | 2.00 |
| Measured FSR (nm) | ~ | 64.5 | 29.8 | 4.4 | 112 |
| Size (mm$^2$) | 9.5 × 16.0 | 4.2 × 13.3 | 17.4 × 22.8 | 30.3 × 32.8 | 11.1 × 20.2 |

[1] The symbol "~" means not applicable.

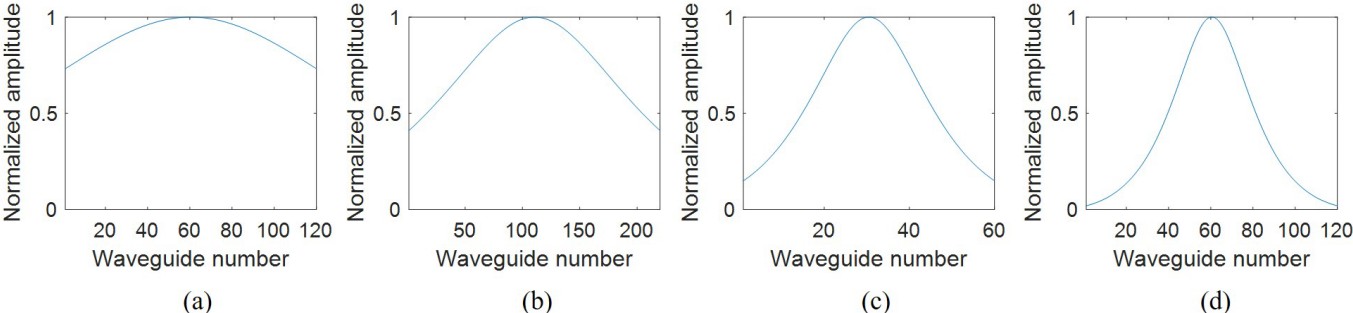

**Figure 4.** Normalized amplitudes at the output facets of the BBAs of (**a**) WSL1, (**b**) WSL2, (**c**) WSL3, and (**d**) WSL4.

### 3.1. Waveguide Lens Only

To demonstrate the focusing function of the waveguide lens, the length of the $i$-th waveguide is designed by setting the grating part ($\Delta L$) to zero in Equation (1). The focal length is set to 10 cm. The simulated far-field distributions of the waveguide lens at the focal plane are shown in Figure 5. The distance between the zero-order and the first-order diffractions increases with longer wavelength, which is consistent with the grating function for multislit diffraction; that is, $d\sin\theta_o = m\lambda$, where $\theta_o$ is the diffraction angle and $m$ is the diffraction order. Higher-order ($m \geq 1$) diffractions will indeed see the dispersion effect, and the diffraction angle also increases for longer wavelength. The zero-order diffraction ($m = 0$) is wavelength independent, meaning that all wavelengths will be focused on the same line. The width of the zero-order focused line, measured as the FWHM of the intensity distribution on the detector plane, is calculated to be 53.5 µm, as shown in Figure 5b.

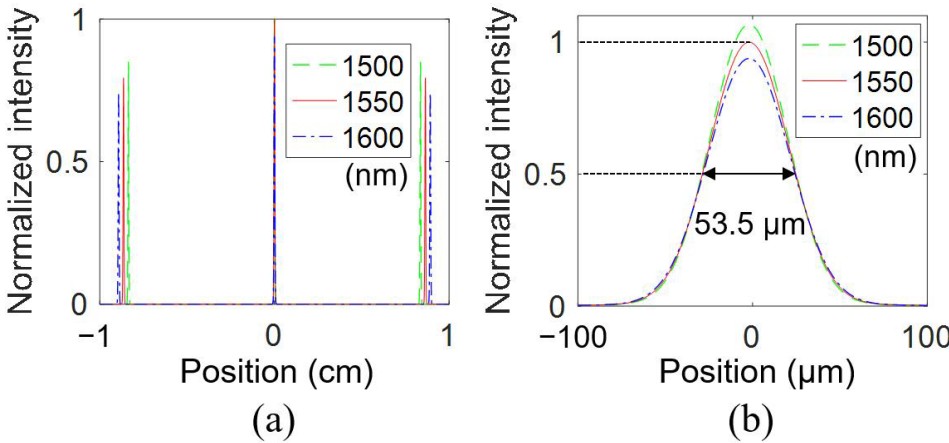

**Figure 5.** (**a**) Simulated far field intensity distributions of the waveguide lens at different wavelengths. (**b**) Zoom-in view of the intensity distributions of the zero-order diffraction at different wavelengths.

### 3.2. WSLs with Different Focal Lengths to Camera

By designing WSLs with different focal lengths, the distance between the output facet of the WSL and the camera can be changed. With a reduced focal length, the capturing loss of the camera can be reduced since more power is received by the camera in the vertical direction. However, as the diffraction angle provided by a WSL is fixed, a short distance to the camera may limit the effective area on the camera sensor; that is, the spectral lines spanning an FSR may cover only a narrow region of the camera sensor. Furthermore, it makes it difficult to reconstruct a continuous spectrum if the spectral lines are suppressed below 1 pixel size. Therefore, the focal length, along with other parameters of the WSL, must be designed according to the requirements from the applications and the specifications of the chosen camera sensor. In this work, we designed WSLs with two different focal lengths, 5 cm (WSL1) and 10 cm (WG lens, WSL2, WSL3, and WSL4), according to Equation (2).

The loss of the spectrometer includes the insertion loss of the WSL chip, the diffraction loss in free space, and the capture loss due to the limited height of the camera sensor [15]. For example, the calculated total insertion losses at 1550 nm of WSL1 from the input fiber to the output facet of the device are 6.14 dB (TE) and 6.04 dB (TM), respectively. Based on the diffraction integral in free space, the diffraction loss due to multiple orders and capture loss due to the limited height of the camera sensor [15] are calculated to be 5.92 dB and 0.97 dB, respectively. The loss optimization methods are discussed in our previous work [15].

### 3.3. WSLs with High Spectral Resolution

Based on Equation (10), if the number of waveguides $N$ is fixed, the FSR and FWHM of the device are proportional to each other. For a WSL with a high resolution, a compromise has to be made with respect to FSR. We have considered this and come up with two WSLs (WSL2 and WSL3) aiming to achieve high spectral resolutions. The two devices have the same focal length. The numbers of waveguides are set as 220 and 60, the FSRs are set as 31 and 4.5 nm, and the simulated FWHMs are 0.14 and 0.09 nm, respectively.

As discussed in Section 2, to calculate the deflection angle of a WSL, the phase difference $\Delta\varphi$ should be normalized in the range of $(-\pi, \pi)$. The normalized $\Delta\varphi$ and deflection angle $\theta$ of WSL2 are calculated according to Equation (6) and plotted in Figure 6 for three FSRs. Under the system implementation as indicated in Figure 1a, within one FSR, the longer wavelengths will appear on the right side of the camera.

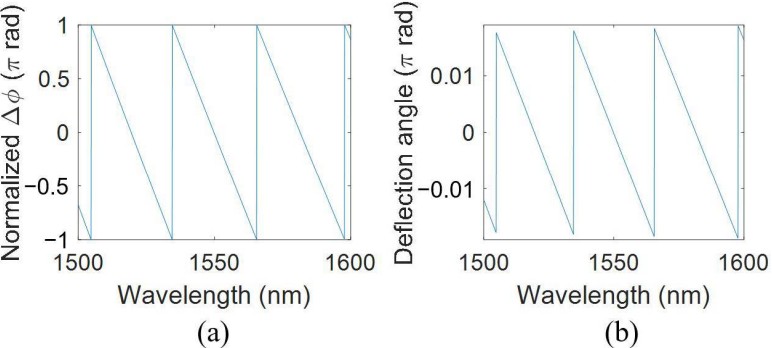

**Figure 6.** (**a**) Normalized phase difference $\Delta\varphi$ and (**b**) deflection angle $\theta$ of WSL2.

### 3.4. WSL with Large FSR

A large FSR can be achieved with a small length difference $\Delta L$, and from the discussion in Section 2, the WSL should adopt the three-arc structure in Figure 3b. We designed WSL4 with an enlarged FSR of 112 nm. The number of waveguides is chosen to be 120, and the resulting FWHM is relaxed to 1.47 nm. For a finer resolution, more waveguides should be included in the design.

## 4. Device Fabrication and Testing Results

### 4.1. Device Fabrication

The designed devices are fabricated on a 4-inch silicon wafer with a standard process as described in our previous work [20]. The core and cladding are polymer materials with refractive indexes of 1.48 and 1.45, respectively. The bottom cladding layer is first spin-coated on the silicon wafer; then a 3 µm thick core layer is spin-coated, and the device is patterned by photolithography and inductively coupled plasma (ICP). After the top cladding is spin-coated and cured, the wafer is diced into bars/chips for characterization with standard sawing equipment. The facets of the chip are not polished.

Figure 7 shows photos of the fabricated devices. Figure 7a shows a photo of WSL2 with the one-arc layout. Figure 7b is a photo of the WG lens without grating effect, and Figure 7c displays WSL4, both adopting the three-arc structure. Some microscope images are placed in the insets, revealing fine structures of the chip. The sizes of the chips are summarized in Table 1.

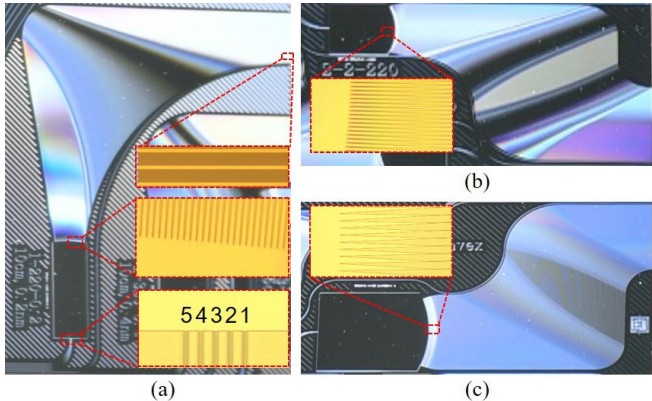

**Figure 7.** Photos of some fabricated devices: (**a**) WSL2, (**b**) WG lens, (**c**) WSL4. The insets are microscope images of the device parts. (5 4 3 2 1) There are five input ports in WSL2.

### 4.2. Experimental Setup

A diagram of the experimental setup for testing the fabricated devices is shown in Figure 8. Light from a tunable laser (EXFO T100S-HP) goes first through a polarization controller (PC) and then is injected to the chip by fiber-chip edge coupling. An infrared

camera (Xenics Bobcat-640-GigE, 640 × 512 pixels, 20 μm pixel pitch) is mounted on a stage and placed at a distance according to the focal length of the chip. The polarization of the input light can be selected by the polarizer inserted between the chip and the camera. As the chosen waveguide platform manifests slight material birefringence, which is verified by the refractive index measurement using a prism coupler, the WSL is essentially polarization dependent, despite the square waveguide core. In this work, the input light is regulated at TE polarization unless otherwise specified. After setting the polarization, the polarizer is removed. The tunable laser and the camera are controlled by a home-developed LabVIEW program. The spectral images are displayed on the computer monitor during the wavelength scan and saved automatically for analysis. The measured parameters of the devices are listed in Table 1.

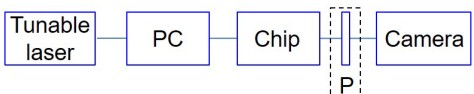

**Figure 8.** Diagram of the chip testing system. PC: polarization controller. P: polarizer.

*4.3. Waveguide Lens*

As shown in Figure 9a, we first demonstrate the near field of a single-mode waveguide fabricated on the same chip of the WG lens, by a collimating lens (Lens1) and a zooming lens (Lens2) between the chip and the camera. The mode profile of a 3 μm × 3 μm waveguide at 1550 nm is shown in Figure 9b. When the free-space lenses are removed, light will diverge, and virtually no patterns can be observed with the same input power, as shown in Figure 9c,d. As a comparison, the output light of the WG lens chip can be focused well onto the camera and form a bright line without any free-space lens, as the wavefront modulation is already included in the waveguide array. The focal length is set as 10 cm. Since the lens function is only implemented on the horizontal plane, light still diverges in the vertical direction, resulting in a line image shown in Figure 9f.

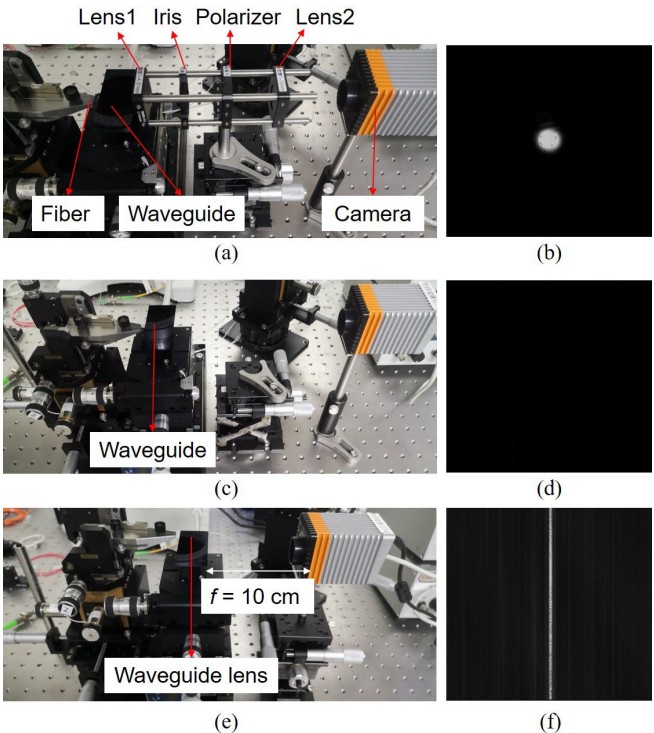

**Figure 9.** Photos of testing systems and captured photos of the optical fields of (**a**,**b**) a single-mode waveguide using free-space lenses, (**c**,**d**) a single-mode waveguide without any free-space lens, and (**e**,**f**) the WG lens chip without any free-space beam control.

### 4.4. WSLs with Different Focal Lengths

By designing the relative waveguide length for the beam focusing function according to Equation (2), the focal length of WSL can be changed. WSL1 is designed with a focal length of 5 cm, whereas the focal lengths of the rest devices in Table 1 are 10 cm. Figure 10a shows the spectral lines of WSL1. Figure 10b shows the measured far-field distributions, which are fitted by multislit diffraction [21]. The fitting method is the same as that in [15]. The FWHM is 0.74 nm. Figure 10c plots the measured and fitted wavelengths at different pixel positions.

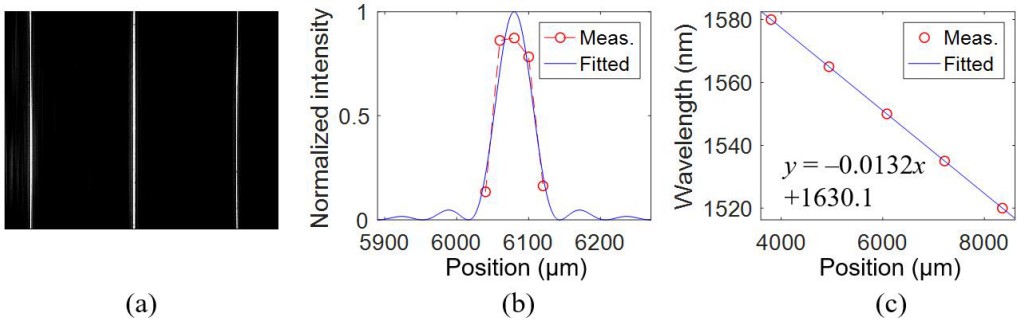

(a)　　　　　　　　　(b)　　　　　　　　　(c)

**Figure 10.** Measured results of WSL1 with a focal length of 5 cm. (**a**) Spectral lines at 1550 nm. (**b**) Measured and fitted far-field distributions with an FWHM of 0.74 nm at 1550 nm. (**c**) Measured and fitted wavelengths at different pixel positions.

### 4.5. WSLs with High Spectral Resolution

The WSL with multiple input waveguides can be used for the simultaneous analysis of signals from multiple input fibers [4]. Additional input waveguide can also be used for calibration purposes [14]. For WSL2, five input ports are arranged at the input facet of the BBA with a spacing of 12 μm, as shown in the inset of Figure 7a. The measured results of WSL2 when input from the middle input port (Port3) are shown in Figure 11. The FWHM is 0.16 nm at 1540 nm. As plotted in Figure 12a, when the input port moves from the right part to the left part of the BBA (i.e., from Port1 to Port5), the position of the spectral line shifts to the right side of the camera. Since the long waveguides are connected to the left part of the BBA, when inputting from the ports on the left side of the BBA, the optical path lengths from the input port to the starting point of the long waveguides will decrease. As a result, the phase differences from the input port to different output waveguide facets are reduced, corresponding to a smaller deflection angle. Therefore, when the input port moves from Port1 to Port5, the spectral line is shifted to the right side of the camera, corresponding to the side of the short waveguides.

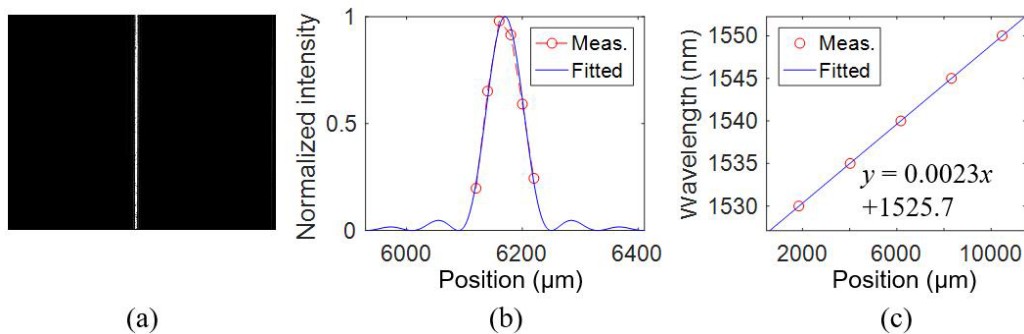

(a)　　　　　　　　　(b)　　　　　　　　　(c)

**Figure 11.** Measured results of WSL2. (**a**) Spectral line at 1540 nm. (**b**) Measured and fitted far-field distributions with an FWHM of 0.16 nm at 1540 nm. (**c**) Measured and fitted wavelengths at different pixel positions.



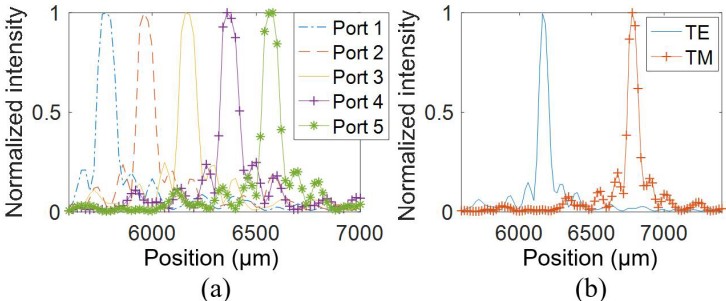

**Figure 12.** Far-field distributions of WSL2 at 1540 nm for (**a**) different input ports under TE polarization and (**b**) the central input Port3 under both polarizations.

To investigate the polarization dependence of the WSL, the polarization of the input light is changed by a polarization controller. As shown in Figure 12b, the spectral line of the TM light is on the right side of the TE light since the refractive indexes of TM are smaller than that of TE for the polymer materials, which means that the TM light has a smaller deflection angle according to Equation (6). The refractive indexes of the core and cladding polymer materials are measured by a prism coupler. The refractive indexes of the core material are 1.47989 (TE) and 1.47891 (TM) at 1550 nm. The refractive indexes of the cladding material are 1.44972 (TE) and 1.44836 (TM) at 1550 nm. For WSL2, the distance between the spectral lines of TE and TM polarizations at 1540 nm is calculated to be 433 μm based on the measured refractive indexes. The measured distance of the spectral lines in Figure 12b is 620 μm. The nonideal cross section and residual strain in the chip may also have an impact on the birefringence of the waveguides. It is worth noting that the polarizer is not necessary to determine the polarization state of the light. We can simply adjust the PC until only the left set of spectral lines stays in the image, and then we are certain that the input light is regulated to TE polarization.

To reach a higher resolution, WSL3 with only 60 waveguides is fabricated. The measured results are shown in Figure 13. The measured far-field intensity is fitted and shown in Figure 13b. The FWHM is found to be 0.15 nm at 1550 nm, corresponding a wavelength resolving power of ~$10^4$. The difference between the FWHMs of WSL2 and WSL3 is not as large as that in the simulation, which can be attributed to the larger phase error of WSL3 as it has a much larger footprint. Figure 13c shows the wavelengths at different pixel positions. The measured FSR is only 4.4 nm but can be expanded by increasing the number of waveguides while keeping a small FWHM, according to Equation (10). The shallow spectral lines near the bright ones in Figure 13a might be caused by the Fabry–Perot resonance within the chip induced by the reflections at the input and output facets, which can be suppressed by antireflection coating at the facets.

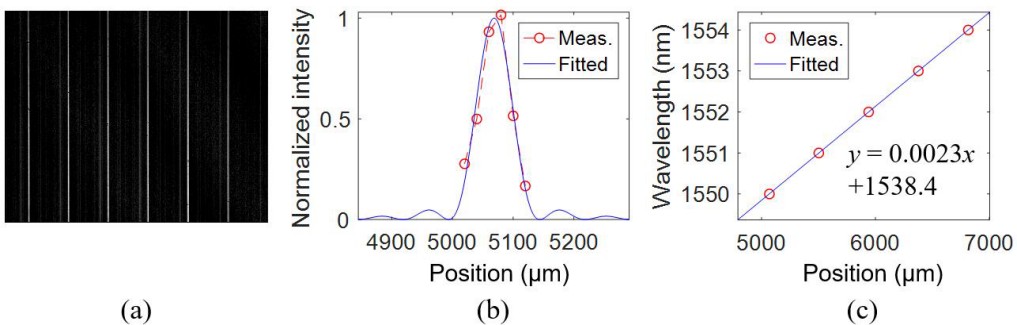

**Figure 13.** Measured results of WSL3. (**a**) Spectral lines at 1550 nm. (**b**) Measured and fitted far-field distributions with an FWHM of 0.15 nm at 1550 nm. (**c**) Measured and fitted wavelengths at different pixel positions.

### 4.6. WSL with Large FSR

The measured results for the device WSL4, targeted at a large FSR, are shown in Figure 14. The measured FSR is 112 nm, and the FWHM is 2 nm. The obtained FWHM is noticeably higher than the calculated value of 1.47 nm. One possible reason is the deviation of the fabricated waveguide geometric parameters from the optimal design, as more bends are present for the three-arc structure. A thorough investigation of phase errors on the broadening of spectral lines will be performed in our future work.

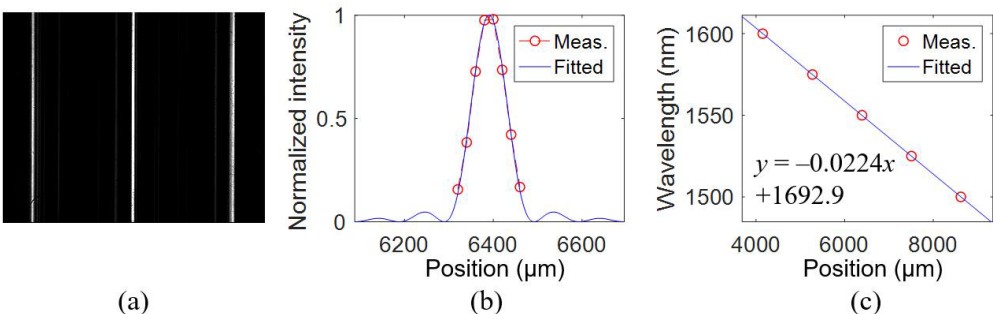

**Figure 14.** Measured results of WSL4. (**a**) Spectral lines at 1550 nm. (**b**) Measured and fitted far-field distributions with an FWHM of 2.0 nm at 1550 nm. (**c**) Measured and fitted wavelengths at different pixel positions.

### 4.7. Measured Spectra of a Multiwavelength Laser

To demonstrate the spectrum measurement function of the WSL in practice, the optical spectrum of an MWL from a handheld multimeter apparatus is measured by WSL2. The result is compared with the spectrum measured by a commercial OSA (Yokogawa AQ6370C).

A diagram of the experimental setup is shown in Figure 15a. The WSL is first calibrated by the tunable laser (EXFO T100S-HP). Then the source is changed to the MWL. Both the polarizations of the tunable laser and the MWL are set to TE. A 1 × 2 splitter is used to split light to the WSL and OSA. When the MWL is selected, the spectral lines are captured by the camera and measured by the OSA simultaneously. A photo of the experimental setup for measuring the optical spectra is shown in Figure 15b.

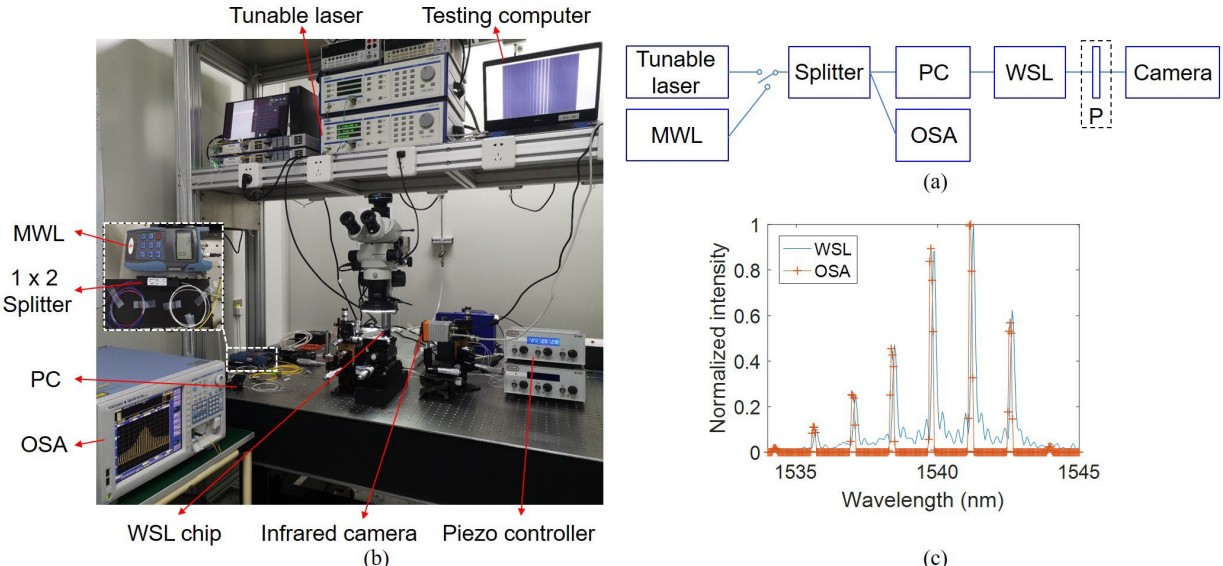

**Figure 15.** (**a**) Diagram and (**b**) photo of the experimental setup for measuring the optical spectra of an MWL. (**c**) Optical spectra of an MWL measured by WSL2 and a commercial OSA. MWL: multiwavelength laser. PC: polarization controller. OSA: optical spectral analyzer.

The measured optical spectra of the MWL are plotted in Figure 15c, and the two methods are in good agreement. The slight red shift of ~0.10 nm by WSL2 might be caused by the calibration error of the WSL. The ripples between the spectral lines measured by WSL2 can be attributed to the phase error of the waveguides from fabrication imperfection.

## 5. Mounting the WSL Chip to a Camera

Since the focal depth of a WSL chip is on the millimeter scale, as listed in Table 1, the WSL chip can be directly mounted to a commercial camera with relaxed alignment precision. An adjustable tube is designed, as shown in Figure 16a. Two axial grooves are fabricated on the inner circle of the tube for holding a metal board with a fiber and a WSL chip on it. The metal board can be moved along the grooves. A screw is used to fix the metal board in the tube. The tube has a C-mount threading, which can be connected to the camera. A locking ring is used to secure the connection once the chip is horizontal, as shown in Figure 16b. A lid is added as a protection cover.

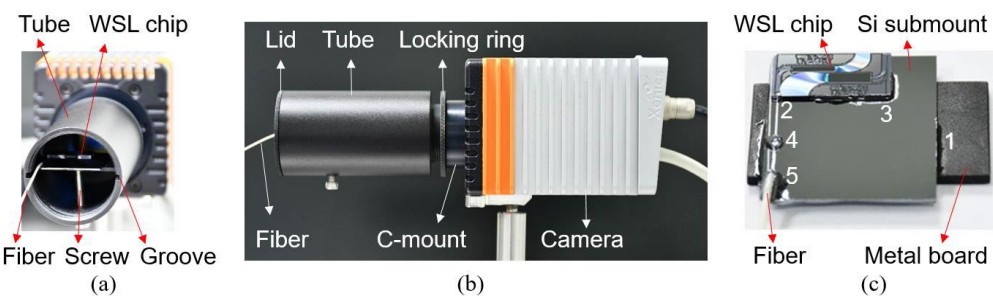

**Figure 16.** (**a**) Inside view of a tube holding a WSL chip. (**b**) Side view of the tube mounted to a C-mount camera. (**c**) Photo of the WSL chip fixed on a silicon submount and a metal board. The labeled numbers indicate the order of UV curing.

For the assembly, a silicon submount is fixed on a metal board. The input fiber is inserted through the central hole of the lid, then aligned with the WSL chip on the fiber-chip coupling system and fixed by an index-matched UV glue. The WSL chip, the bare fiber, and the fiber socket are fixed on the silicon submount successively. A photo of the WSL chip on a silicon submount and a metal board is shown in Figure 16c, where the labeled numbers indicate the order of UV curing. After curing, the tube is connected to the camera. The metal board is then inserted into the tube and stops at the designed position according to the focal length. The metal board is fixed by the screw on the tube. After that, the tube is rotated to keep the chip horizontal, and the locking ring is fixed. Finally, the tube is covered with the lid, as shown in Figure 16b. As described above, once the input fiber is fixed on the chip, no fine alignment is needed for mounting the chip to the camera. The fiber-chip attachment can also be performed passively via mature technology (e.g., on-chip U-grooves [9] and V-grooves [22]).

Figure 17a shows the experimental setup for testing the tube spectrometer with WSL1 inside. The polarization of the light can be judged by the relative positions of the TE and TM spectral lines. The captured spectral lines and the far-field distribution under TE polarization are shown in Figure 17b,c. The measured FWHM is 0.73 nm, which is similar to that measured on the fiber-chip coupling system.

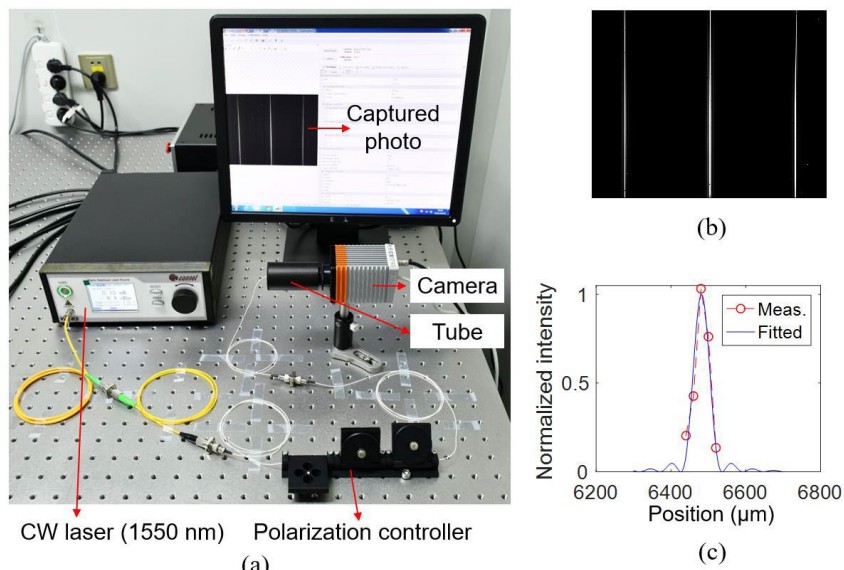

**Figure 17.** (**a**) Experimental setup for testing the WSL chip mounted to the camera. (**b**) Spectral lines at 1550 nm. (**c**) Measured and fitted far-field distributions with an FWHM of 0.73 nm at 1550 nm.

## 6. Conclusions

In this paper, the design, fabrication, testing, and mounting of WSLs have been investigated. Light from a single-mode fiber can be dispersed and focused on a camera at a designed distance. By designing one-arc- or three-arc-based geometric structures, WSLs with high resolution or large FSR can be realized, respectively. In the experiment, WSLs with different focal lengths, FWHMs, and FSRs are demonstrated. The spectrum of a handheld laser source is measured by a WSL, and the result is in good agreement with a commercial OSA. For practical applications, a tube-based low-cost packaging scheme is demonstrated, capable of mounting to a standard camera without any extra optical elements. We believe this device may have applications in optical fiber sensing and other research fields, thanks to its compact footprint, flexible design, and simple mounting technique to a camera without using free-space optical elements or customized electronics. As future work, we will investigate the FWHM and FSR limits in consideration of the fabrication-induced phase error and the wavelength dependence of the waveguide effective index. Furthermore, multilayer WSLs can be designed to obtain a 2D spectral spot map instead of spectral lines for a more efficient use of the camera sensor area. On the application side, we plan to implement the WSL technology for the development of handheld devices for fast on-site interrogation of spectrum-based optical fiber sensors.

**Author Contributions:** Conceptualization, Z.Z. and X.J.; methodology, X.J. and Z.Z.; software, X.J.; validation, X.J.; formal analysis, X.J. and Z.Z.; investigation, X.J.; resources, Z.Z.; data curation, X.J.; writing—original draft preparation, X.J. and Z.Z.; writing—review and editing, X.J. and Z.Z.; visualization, X.J.; supervision, Z.Z.; project administration, Z.Z.; funding acquisition, X.J. and Z.Z. All authors have read and agreed to the published version of the manuscript.

**Funding:** This research was funded by the National Natural Science Foundation of China, grant number 61905202.

**Institutional Review Board Statement:** Not applicable.

**Informed Consent Statement:** Not applicable.

**Data Availability Statement:** The data presented in this study are available from the corresponding author upon reasonable request.

**Conflicts of Interest:** The authors declare no conflict of interest.

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
