# Peer review of "Planar Waveguide-Based Fiber Spectrum Analyzer Mountable to Commercial Camera"

_photonics, doi:10.3390/photonics9070456_

Round 1

Reviewer 1 Report

This work presents a device referred to as waveguide spectral lens. With an array of on-chip optical waveguides, this device separates the wavelength components from an input port and focuses them onto corresponding positions on a focal plane and is detected using a camera. This is enabled by careful engineering of both the linear and quadratic propagation phases for the waveguide array. This work presents design methodology of their device geometry, numerical calculations, optical measurements, and a demonstration of the device as an optical spectrometer. While arrayed waveguide gratings are well-established in the literature, in my view, this device presents an interesting combination of the array waveguide grating and a free-space detection scheme. This device therefore inherits some advantages and disadvantages of both systems. There seems to be some overlap between this work and Ref. [15] cited in this work, including the phase curvature engineering and the one-arc device design. This work develops the system further and proposes some applications.

               The authors provide detailed derivation for their design methodology in Sec. 2 in Eq. 14-28. These equations correspond the geometry parameters in Fig. 3b to the constrains of the system, such as phase profile for each waveguide and the routing requirements for positioning the waveguide ends on the emission edge. For better readability of this work, I would suggest the authors describe the number of degrees of freedom in Fig. 3b and correspond them to the number of constrains to be satisfied at the beginning of this section. In my view, moving the detailed derivation in Eq. 14-28 to a separate section or supplementary materials may facilitate a smoother reading experience for future readers.

               This work presents calculation and measurements for the spectral ranges and resolution of the device in question. Without doubt, these are critical characteristics for a spectrometer. However, the power efficiency is also an important factor as it could impact the sensitivity and signal-to-noise ratio on a detector. The authors should provide optical power information for such as the input optical power in the laser measurements in Fig. 15c, or perhaps normalizing the intensity plots in Fig 10b to the total input power instead of the peak intensity of the resulting line. In my view, discussion of the power requirements and efficiencies will help strengthen the paper.

               The waveguide array focuses light in the plane of the chip, but creates a diverging beam on the out-of-plane direction due to the small vertical size of the waveguide mode profiles. This seems to be a limitation for the hybrid system presented in this work. In comparison, a fully integrated array waveguide grating system with on-chip detectors will likely not suffer this issue, while a system using a macroscopic lens is capable of focusing light in the out-of-plane direction as well. Could the authors comment on whether this is a fundamental limitation to the system, and if so, does this compromise the claimed advantages of this hybrid system in comparison to the fully on-chip or free-space lens systems? Some discussion on the impact of the beam divergence on detection efficiency at the designed focal length and expected detector sizes would be helpful.

The following are a few other minor issues I would like to bright to the authors’ attention:

The parameters xi, yi are mentioned in Line 155. Please consider denoting xi, yi on Fig 3c, or at least mark out the termination / emission edge on Fig. 3c.

Line 222 describes the waveguide spacing of the 12um, while the operation wavelength is only mentioned until Line 249. Consider comparing the waveguide spacing to the wavelengths or the evanescent field 1/e^2 length when it is first described.

Fig. 4 compares the amplitude profiles of various devices. However, the y axis scales are different (for example Panel a is cut off at 0.7 of the peak amplitude, while Panel d goes all the way to 0). Consider linking the y scales to facilitate comparison. Additionally, would it be helpful to plot the x axis as distance instead of waveguide number?

Line 351 claims that the birefringence of the waveguides resulted from ‘material birefringence’, could the authors describe the rationalization of why the polymer material, which presumably is amorphous, would demonstrate material birefringence? Is there sufficient evidence to rule out other causes, such as geometry (slight differences in waveguide cross-section from square) or residual strain in the material?

Fig. 10, 11, 13, and 14 Panel b show side-lobe structures on the fitted curves but without measured data points in the side-lobe. Would a simple Lorentzian fit suffice for these fits to extract information such as FWHM?

Line 393 describes the device ‘WSL2’ has several input waveguides. Could the authors elaborate on the motivation for this design? Is it meant to widen the measurable spectral range?

Author Response

We would like to thank the reviewer for the comments on our manuscript entitled “Planar waveguide-based fiber spectrum analyzer mountable to commercial camera”. The manuscript has been revised accordingly. Here we provide this response letter with the reviewer’s comments in italic, our replies are styled bold, and with our revisions underlined.

Reviewer 2 Report

The authors provided a detailed study of an AWG based spectrometer design. The manuscript covers the whole process of device design to fabrication and testing. The results are interesting and sound. I have some questions for the authors before recommending the manuscript for publication:

1.     What is the coupling efficiency of the spectrometer? How much power can reach the detect assuming 1 milliwatts input power. If the free space light has pretty complicated mode profile will the efficiency drop vastly?

2.     Will there be any differences when the input light is incoherent? The author should provide some test experiments with incoherent light, such as PL, tungsten light or sunlight.

3.     Some figures have very abbreviated labels and should be replaced with a more clear description. For example “D”. and Normalized value usually comes with (a.u.) to indicate the arbitrary units.

Author Response

(The authors gave the same response as above.)

Reviewer 3 Report

Thank you for the well prepared and very interesting paper. It was a pleasure to read it. The engineering approach for the waveguide spectral lenses is very clearly described and thus may serve as a convenient starting point for applications or further research. The examples are well chosen and described and the results resemble the simulations to a reasonable amount.

I found just two minor typos:

Abstract, line 9: "spectra" should be "spectral"

4.2, line 347: "injects" should be "is injected"

Author Response

(The authors gave the same response as above.)
